# Distributed Architecture Search Over Heterogeneous Distributions

**Erum Mushtaq**                                                    *emushtaq@usc.edu*
*Department of Electrical and Computer Engineering*
*University of Southern California*

**Chaoyang He**[*]                                                    *ch@fedml.ai*
*FedML, Inc*

**Jie Ding**                                                         *dingj@umn.edu*
*School of Statistics*
*University of Minnesota*

**Salman Avestimehr**                                               *avestime@usc.edu*
*Department of Electrical and Computer Engineering*
*University of Southern California*

**Reviewed on OpenReview:** *https://openreview.net/forum?id=sY75NqDRk1*

## Abstract

Federated learning (FL) is an efficient learning framework that assists distributed machine learning when data cannot be shared with a centralized server. Recent advancements in FL use predefined architecture-based learning for all clients. However, given that clients' data are invisible to the server and data distributions are non-identical across clients, a predefined architecture discovered in a centralized setting may not be an optimal solution for all the clients in FL. Motivated by this challenge, we introduce SPIDER, an algorithmic framework that aims to **S**earch **P**ersonal**I**zed neural architecture for fe**DER**ated learning. SPIDER is designed based on two unique features: (1) alternately optimizing one architecture-homogeneous global model in a generic FL manner and architecture-heterogeneous local models that are connected to the global model by weight-sharing-based regularization, (2) achieving architecture-heterogeneous local models by a perturbation-based neural architecture search method. Experimental results demonstrate superior prediction performance compared with other state-of-the-art personalization methods. Code is available at https://github.com/ErumMushtaq/SPIDER.git .

## 1 Introduction

Federated Learning (FL) is a promising distributed machine learning framework that facilitates data privacy and low communication costs. It has been extensively explored in various machine learning domains such as computer vision, natural language processing, and data mining. Despite many benefits of FL, one major challenge involved in FL is data heterogeneity, meaning that the data distributions across clients are not identically or independently distributed (non-I.I.D). The non-I.I.D distributions result in the varying performance of a globally learned model across different clients. In addition to data heterogeneity, data invisibility is another challenge in FL. Since clients' private data remain invisible to the server, from the server's perspective, it is unclear how to select a pre-defined architecture from a pool of all available candidates (Ding et al., 2018). In practice, it may require extensive experiments and hyper-parameter tuning over different architectures, a procedure that can be prohibitively expensive.

---

[*]This work was done while Chaoyang He was a PhD student at USC.

The current literature on this subject addresses the data-heterogeneity challenge by exploring the variants of the standard FedAvg to train a global model, including the `FedProx` (Li et al., 2018), `FedOPT` (Reddi et al., 2021), and `FedNova` (Wang et al., 2020). In addition to training a global model, frameworks that focus on training personalized models have also gained a lot of popularity. The `PerFedAvg` (Fallah et al., 2020), `pFedMe` (Dinh et al., 2020), `Ditto` (Li et al., 2021), and `SelfFL` (Chen et al., 2022) are some of the recent works that have shown promising results to obtain improved performance across clients. However, all these works exploit pre-defined architectures and adapt the optimization algorithm to accommodate data heterogeneity. Consequently, in addition to their inherent hyper-parameter tuning, these personalization frameworks often encounter the invisibility of training data challenge (Majeed et al., 2022) that one has to select a suitable model architecture involving a lot of hyper-parameter tuning.

There is another line of work that has recently emerged that uses neural architecture search (NAS) in federated learning. In FL, NAS has been explored for a `global architecture search` (He et al., 2020b; Yuan et al., 2022), `partial model personalization` (Hoang & Kingsford, 2021), `cluster-based architecture search` (Wang et al., 2022) and `resource-aware personalization` (Dudziak et al., 2022; Isik et al., 2023). However, we argue that a `global architecture search` may not capture the data heterogeneity in FL with non-I.I.D data. Additionally, `partial model personalization method` searches some components of a predefined model, and therefore, needs a search in order to identify an optimal boundary between personalized and shared (global) components for each client. This search can become prohibitively expensive if not conducted in an automated fashion. The `cluster-based architecture search` methods search the same architecture per cluster, often requiring clients to share their data distribution information to form clusters, which may not be feasible for clients such as hospitals due to privacy concerns.

Alternatively, we address the data heterogeneity challenge by facilitating each client to search the entire architecture space and personalize architecture to each local client's data distribution in a non-cluster-based setting. Oftentimes, the search for an optimal architecture is regarded as a hyper-parameter tuning problem for a given task (or a data distribution) (Kim et al., 2021). Since data is non-I.I.D due to different data-generative models at different silos, the selection of the optimal hyper-parameters/architectures can be different across different silos. Therefore, in such a setting, we propose to enable each silo to search for a personalized architecture and capture its own data distribution.

To search personalized architectures, we introduce SPIDER, an algorithmic framework that aims to **S**earch **P**ersonal**I**zed neural architecture for each individual client in fe**DER**ated learning. Particularly, we focus on the cross-silo setting where clients have ample computational resources but data heterogeneity is a main concern Huang et al. (2022). Under this setting, SPIDER deploys two models, a local and a global model, on each client. Initially, both models use the DARTS search-space-based Supernet (Liu et al., 2019), an over-parameterized architecture search space. The global model is shared with the server for the FL updates and, therefore, stays the same in the architecture design. On the other hand, the local model performs a personalized neural architecture search and gets updated. The searched local/child models can be heterogeneous across silos. To search and optimize the heterogeneous local models while benefiting from the global model, we exploit SPIDER Trainer which 1) trains the local and global models in an alternate fashion and 2) regularizes the common connections between the global model's Supernet and the local model's child model. This regularization essentially distills knowledge from the global model to the local model. The proposed approach not only aids in uncovering different personalized architectures across various silos but also ensures that the derived local model remains local, with no sharing to the server and other clients.

To evaluate the performance of the proposed algorithm, we consider a cross-silo FL setting and use the Dirichlet Distribution to create a non-I.I.D data distribution across clients. For evaluation, we report the test accuracy at each client using the 20% of training data kept as test data for that client. Furthermore, we compare our work to the state-of-the-art predefined architecture-based personalization FL schemes. We demonstrate that the architecture personalization yields better results than state-of-the-art personalization algorithms based solely on the optimization layer, such as Ditto (Li et al., 2021), perFedAvg (Fallah et al., 2020), local adaptation (Cheng et al., 2021) and KNN-Per (Marfoq et al., 2022). Additionally, we also compare our work to a cluster-based hypernetwork search method, FedMN (Wang et al., 2022).

To summarize, the key contributions of our work are as follows:

• Personalized Architecture Search: We propose and formulate a personalized neural architecture search framework called SPIDER for personalized federated learning (FL). Architecture personalization allows each silo to capture the complexities of its local data distribution at the level of architecture design and potentially leads to improved personalization.

• Architecture-Heterogeneous Model Training: SPIDER formulates architecture personalization as a bilevel optimization. It solves the bilevel optimization objective by incorporating two distinct features: 1) exploiting an alternate training-based SPIDER Trainer that trains one architecture-homogeneous global model, also called Supernet, in a generic FL manner and one architecture-heterogeneous local model that is connected to the global model by weight-sharing-based regularization, 2) searching architecture-heterogeneous local model by performing neural architecture search locally. The proposed approach facilitates knowledge sharing from the global model to the local model, enabling effective learning from other silos in the presence of heterogeneous personalized architecture settings.

• Performance Evaluation: We conducted extensive experiments to demonstrate the benefits of SPIDER compared to state-of-the-art personalized federated learning approaches, including `Ditto` (Li et al., 2021), `perFedAvg` (Fallah et al., 2020), `Local Adaptation` (Cheng et al., 2021), `KNN-Per` (Marfoq et al., 2022), and `FedMN` (Wang et al., 2022). We observed significant improvements in average local accuracy compared to the aforementioned approaches on three datasets: CIFAR10, CIFAR100, and CINIC10 using the ResNet18 model. Among the state-of-the-art methods, KNN-Per obtains the highest accuracy for all three datasets. SPIDER outperforms this PFL baseline by an accuracy margin of 3.29%, 5.31%, and 1.59% on CIFAR10, CIFAR100, and CINIC10 datasets, respectively. To assess the gain achieved through architecture personalization, we measure the accuracy gap between the current client's model and models attained from other clients and fine-tuned on the current client's data. On average, we observed an accuracy drop with the other clients' fine-tuned model compared to its own searched architecture. We refer to this accuracy drop as the "personalization gain" achieved by architecture personalization. Averaged over all clients, we observed personalization gains of 4.70%, 7.84%, and 4.19% on CIFAR10, CIFAR100, and CINIC10 datasets, respectively.

## 2    Related works

**Heterogeneous Architecture for FL**    Heterogeneous neural architecture has primarily been explored to personalize models for system/hardware heterogeneity in cross-device FL scenarios. For instance, the work by Lin et al. (2020) accomplishes the task of heterogeneous model aggregation by forming clusters of clients assigned a predefined model and allowing for heterogeneous models across clusters. Model aggregation is based on cluster-wise aggregation followed by a knowledge distillation from the aggregated models into the global model. Another work, HeteroFL (Diao et al., 2021), aggregates heterogeneous models by assigning static sub-parts of the global model based on computation budgets and aggregating the common parameters across different clients. Similarly, in the work (Luo et al., 2021), a limited channel-wise search is performed to assign sub-models that meet clients' efficiency budgets, followed by partial weight aggregation at the server. These random or static partial heterogeneous model aggregation schemes are susceptible to client drifts, which arise from inconsistencies between local model architectures and the global model architecture. FedRolex (Alam et al., 2022) is a recent approach that addresses client drifts by allowing the training of sub-models extracted via a rolling window from a large pre-defined global model. However, it cannot maintain static heterogeneous local models across silos. Additionally, all these works primarily focus on addressing system heterogeneity challenges in FL by considering the computing capabilities of training devices. The sub-models are extracted randomly, statically, or based on a rolling window. In FL, where data visibility is limited, determining which architecture would be suitable for each client based on their data distribution is a challenging task that requires exploration. Our proposed method focuses on tailoring architecture to individual clients' data distributions by searching for model-heterogeneous architectures across silos. By prioritizing personalized architecture design, we aim to address the challenge of data distribution variations and enable more effective model customization in FL settings.

**Neural Architecture Search for FL**    Neural Architecture Search (NAS) has gained momentum in recent literature for searching a global model in a federated setting. FedNAS (He et al., 2020b) explores the MileNAS

solver (He et al., 2020d) with the Federated averaging algorithm (McMahan et al., 2017) to search for a global model. Direct Federated NAS (Garg et al., 2020) investigates the compatibility of DSNAS (Hu et al., 2020) with Federated learning for searching a global model. Work (Zhu & Jin, 2021) utilizes evolutionary NAS to design a global model. Work (Singh et al., 2020) explores differential privacy using the DARTs solver (Liu et al., 2019) to examine the trade-off between accuracy and privacy in a global model. Work (Xu et al., 2020) start with a pre-trained handcrafted model and continue to prune the model until it meets the efficiency budget. FedPNAS (Hoang & Kingsford, 2021) divide the model architecture into global and personal components and search for the personal component for personalization on identical and independent (IID) vision tasks. However, this approach remains unexplored in the non-I.I.D data distribution settings of FL. FedPM (Isik et al., 2023) addresses the communication efficiency challenge by learning a stochastic binary mask of a dense network with fixed weights. FedorAS (Dudziak et al., 2022) addresses the system heterogeneity challenge in a cross-device setting by searching architectures in cluster/tier-based settings to meet an efficiency budget. FedMN (Wang et al., 2022) proposes a cluster-based module personalization approach. Cluster-based approaches often require information exchange regarding data distributions to form clusters, which may not be feasible due to privacy concerns. This paper aims to search for an entire personalized neural architecture for each client that remains unknown on the server.

## 3 Preliminaries, Motivation, and Design Goals

In this section, we introduce the state-of-the-art methods for personalized federated learning, discuss the motivation for personalizing model architectures, and summarize our design goals.

**Personalized Federated Learning**  A natural formulation of FL is to assume that among $c$ distinct clients, each client $k$ has its own distribution $P_k$, draws data samples from $P_k$, and aims to solve a supervised learning task (e.g., image classification) by optimizing a global model $\boldsymbol{w}$ with other clients collaboratively. At a high-level abstraction, the optimization objective is then defined as:

$$\min_{\boldsymbol{w}} G\left(F_1(\boldsymbol{w}, \mathcal{A}), ..., F_c(\boldsymbol{w}, \mathcal{A})\right), \tag{1}$$

where $F_k(\boldsymbol{w}, \mathcal{A})$ is client $k$'s local objective function that optimizes the weight parameters $\boldsymbol{w}$ of the global model $\mathcal{A}$ and $G$ is the global model aggregation function that aggregates each client's local objectives. For example, for FedAvg (McMahan et al., 2017), $G(.)$ would be weighted aggregation of the local objectives, $\sum_{k=1}^{c} p_k F_k(\boldsymbol{w}, \mathcal{A})$, where $\sum_{k=1}^{c} p_k = 1$. However, as distributions across individual clients are typically heterogeneous (i.e., non-I.I.D.), there is a growing line of research that advocates reformulating FL as a personalization framework, dubbed as personalized FL (PFL). In PFL, the objective is redirected to train a personalized model $\boldsymbol{v}_k$ for device $k$ that performs well on the local data distribution while also learning from other silos, such as via aggregation function $G$.

To solve this challenging problem, various PFL methods are proposed, including FedAvg with local adaptation (Local-FL) (Cheng et al., 2021), MAML-based PFL (MAML-FL) (Fallah et al., 2020; Jiang et al., 2019), clustered FL (CFL) (Ghosh et al., 2020; Sattler et al., 2021; Wang et al., 2022; Dudziak et al., 2022), (Liang et al., 2019; Pillutla et al., 2022), federated multitask learning (FMTL) (Smith et al., 2017) and knowledge distillation (KD) (Lin et al., 2020; He et al., 2020a). Additionally, Neural Architecture Search (NAS) has gained momentum in recent literature to search for a personalized model in a federated setting. FEDPNAS (Hoang & Kingsford, 2021), personalized layer-based FL (PL-FL), divides the model architecture into global and personal components and searches the personal component's architecture for personalization. Further, FedMN (Wang et al., 2022) follow clustered FL (CFL) approach to address the data heterogeneity challenge in FL.

**Motivation for Neural Architecture Personalization**  Distinct from these existing works on PFL, we propose a new approach to instead personalize the entire model architecture for each client. We are motivated by one critical potential benefit, that is, the searched architecture for each client is expected to fit its own distinct distribution, which has the potential to provide a substantial improvement over the existing PFL baselines that only personalize model weights. In addition, a personalized architecture search allows

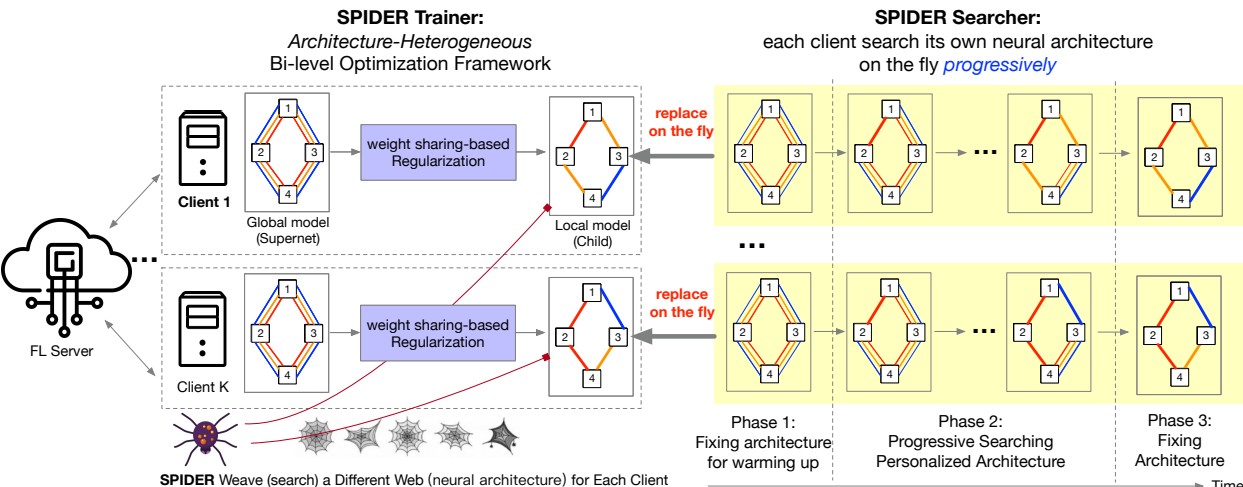

*Figure 1: Illustration of SPIDER framework. SPIDER consists of two components: Trainer (global and local model parameter optimization) and Searcher (local architecture search). Searcher weaves (searches) a different web (neural architecture) for each client in three phases: Phase 1 (Local Architecture Pre-training), Phase 2 (Local Architecture Search), Phase 3 (Local Architecture Training). To illustrate the local architecture search, we represent a simplified cell with 4 nodes where each node represents a latent feature map and each edge is associated with 3 operations (blue, orange ad yellow). We provide details about the cells of DARTS architecture search space in Section 4.2 and Appendix B.2 and show the heterogeneous client architecture cells obtained through the search process in Figure 2.*

the clients to even keep their local model architectures private in a sense the server and other clients neither knows the architecture nor the weights of that architecture.

**Design Goals** Our goal is to enable a complete model's personalized neural architecture search for all clients in FL. In this context, the limitation of existing personalized FL methods is obvious: Local-FL and MAML-FL need every client to have the same architecture to perform local adaptations; In CFL, the clustering step requires all clients in one cluster to share a homogeneous model architecture; PL-FL can only obtain heterogeneous architectures for a small portion of personalized layers, but it does not provide an architecture-agnostic method to determine the boundary of personalized layers in an automated mechanism; FMTL is a regularization-based method which cannot perform regularization when architectures are heterogeneous across clients; KD has an unrealistic assumption that the server has public dataset as the auxiliary data for knowledge distillation.

To circumvent these limitations, our goal is to design an architecture-personalized FL framework with the following requirements:

- **R1**: *Allowing heterogeneous architectures for all clients, which can capture data heterogeneity;*

- **R2**: *Searching and personalizing the entire architecture space to avoid heuristic hyper-parameter search traditionally required to determine the personalized layers and globally shared layers;*

- **R3**: *Requiring no auxiliary data at the client- or server-side (unlike knowledge distillation-based PFL approaches);*

We now introduce SPIDER which meets the above requirements in a unified framework.

## 4 Methodology: SPIDER

### 4.1 Overview

The overall framework of SPIDER is illustrated in Figure 1. In this framework, each client maintains two models: a homogeneous global model for collaborative training with other clients, and a heterogeneous local

model that initially shares the same super architecture space as the global model. SPIDER is formulated as an architecture-personalized bi-level optimization problem (Section 4.2). It consists of two main components: **SPIDER Trainer** (Section 4.3) and **SPIDER-Searcher** (Section 4.4). **SPIDER Trainer** is an architecture-personalized training framework that enables the collaborative training of heterogeneous neural architectures across clients. It utilizes alternate bi-level optimization and weight sharing between personalized architectures and the global model to facilitate federated training. **SPIDER-Searcher**, on the other hand, is designed to dynamically adjust the architecture of each client's local model. It uses a novel neural architecture search (NAS) method that progressively searches for optimal local subnets by perturbing operations at the accuracy level. This allows for personalized architecture search based on each client's local data distribution. Each client's local model goes through three phases: pre-training to warm up the initial model, progressive neural architecture search, and final training of the searched architecture-personalized model. These phases are depicted in Figure 1.

SPIDER can meet the design goals **R1**-**R3** introduced in Section 3 because 1) each client performs independent architecture personalization using its own private data (**R1**), 2) the SPIDER's search space is not restricted to a portion of the model (**R2**), and 3) no auxiliary data is used to assist the search/train process (**R3**).

## 4.2 SPIDER Formulation: Architecture-personalized Bi-level Optimization

SPIDER aims to personalize (weave) a distinct neural architecture (web) for each client. To accomplish this, SPIDER leverages two models, namely a local model $\mathcal{A}_k$ and a global model $\mathcal{A}$, at each client $k \in \{1, 2, ..., c\}$. The two-model design has been shown effective in personalized federated learning (Li et al., 2021; Chen et al., 2022). However, the prior works only target predefined architectures. Our work proposes to facilitate personalization at the architectural level by neural architecture search (NAS). To achieve this, we use DARTS' search space (Liu et al., 2019) for both $\mathcal{A}$ and $\mathcal{A}_k$. Essentially, DARTS' search space consists of repetitions of cell-based microstructures. Each cell can be considered as a directed acyclic graph (DAG) with $N$ nodes and $E$ edges, where each node represents a latent feature map, and each edge can be associated with multiple operations $O$ (e.g. "skip connect", "sep conv 3x3"). For instance, in Figure 1, during Phase 1, the cell is depicted with 4 nodes and 3 operations per edge for the purpose of illustration. At any specific edge $i$, each operation $j$ has its corresponding learnable architecture parameter $\alpha_{ij} \in \{0, 1\}$. More specifically, the output at the edge $i$ is constructed by $x = \sum_{j=0}^{|O|-1} \alpha_{ij} o_j(x^i)$, where $x^i$ is the input at the edge $i$, $o_j(x^i)$ is the output of function $o_j(.)$ (operation index $j$ in the operation candidate search space $O$) applied to the input $x^i$, $\alpha_{ij}$ is the associated architecture parameter and $x$ is the mixed operation output at the edge $i$. We introduce $\mathcal{A}$ and $\mathcal{A}_k$ to be a collection of all $\alpha_{ij}$ and $\alpha_{ij_k}$ for $\forall i, j$, respectively. Initially, all $\alpha_{ij}$ are fixed to 1 for all $i \in E$ and $j \in O$ in $\mathcal{A}$ and $\mathcal{A}_k$. This search space searches an architecture among $4^{14}$ possible architecture configurations for each client. We provide details of the DARTS search space in Appendix B.2.

In most PFL works (Li et al., 2021), the global model architecture and the local model architectures are fixed (pre-defined). However, SPIDER facilitates each client to search for its own local architecture $\mathcal{A}_k$. To achieve this objective, SPIDER formulates a local objective $F_k(\boldsymbol{v}_k, \mathcal{A}_k)$ which aims to perform neural architecture search for $\mathcal{A}_k$ by learning its architecture parameters $\alpha_{ij_k}$ and optimize its corresponding model weights $\boldsymbol{v}_k$. Additionally, it also maintains a global objective to update the global model weights. It connects the local and global objectives with a distance-based regularization between the parameters of the global model and the local model. More specifically, the SPIDER formulation follows an architecture-personalized bi-level optimization problem for each client $k$:

$$\min_{\boldsymbol{v}_k \in \mathbb{R}^{d_k} \subseteq \mathbb{R}^d, \mathcal{A}_k \subseteq \mathcal{A}} h_k(\boldsymbol{v}_k, \mathcal{A}_k; \boldsymbol{w}^*, \mathcal{A}) = F_k(\boldsymbol{v}_k, \mathcal{A}_k) + \frac{\lambda}{2} \operatorname{Reg}(\boldsymbol{v}_k, \boldsymbol{w}^*) \tag{2}$$

$$\text{s.t.} \quad \boldsymbol{w}^* \in \arg\min_{\boldsymbol{w} \in \mathbb{R}^d} G\left(F_1(\boldsymbol{w}, \mathcal{A}), ..., F_K(\boldsymbol{w}, \mathcal{A})\right), \tag{3}$$

where global model architecture $\mathcal{A}$ is parameterized by weight parameters $\boldsymbol{w}$ with size $d$, and local model architecture $\mathcal{A}_k$ is parameterized by weight parameters $\boldsymbol{v}_k$ with parameter size $d_k$ for client $k$. $\operatorname{Reg}(\boldsymbol{v}_k, \boldsymbol{w}^*)$ illustrates a distance-based regularization between the global and local model. We present the local objective in Equation 2 and the global objective in Equation 3. Further, we use a cross-entropy loss function as

our $F_k(.)$ objective function. As shown in Equation 2, the lower-level optimization optimizes the global model parameters $\boldsymbol{w}^*$, and the upper-level optimization optimizes the local architecture $\mathcal{A}_k$ and its weight parameters $\boldsymbol{v}_k$. Here the hyperparameter $\lambda$ controls the interpolation between local model weight parameters $\boldsymbol{v}_k$ and global model weight parameters $\boldsymbol{w}^*$ models. Since $\mathcal{A}_k$ evolves as learning progresses, it becomes a subset of the global model $\mathcal{A}$, denoted as $\mathcal{A}_k \subseteq \mathcal{A}$, and $\mathbb{R}^{d_k} \subseteq \mathbb{R}^d$.

### 4.3 SPIDER Trainer: Federated Training on Heterogeneous Architectures

SPIDER facilitates each silo to search and train an architecture better suited for its specific data distribution. However as the local architecture $\mathcal{A}_k$ evolves due to our proposed formulation, it becomes a subset of the global model $\mathcal{A}$, denoted as $\mathcal{A}_k \subseteq \mathcal{A}$, and $\mathbb{R}^{d_k} \subseteq \mathbb{R}^d$. In such a setting, the challenge of the proposed local optimization objective is to devise the regularization between the architecture-homogeneous global model and architecture-heterogeneous local models. To solve this challenge, we propose SPIDER trainer, an architecture-personalized training framework that can collaboratively train heterogeneous local neural architectures across clients.

To clearly show how SPIDER handles the optimization difficulty of Equation 2, we first downgrade the objective to the case that all clients use *predefined* (fixed) heterogeneous architectures (derived from the Supernet $\mathcal{A}$). More specially, we reduce the aforementioned optimization framework in Equation 2 and Equation 3 to the following:

$$\min_{\boldsymbol{v}_k \in \mathbb{R}^{d_k}} \quad h_k(\boldsymbol{v}_k, \mathcal{A}_k; \boldsymbol{w}^*, \mathcal{A}) = F_k(\boldsymbol{v}_k, \mathcal{A}_k) + \frac{\lambda}{2} ||\boldsymbol{v}_k - \boldsymbol{w}^*_{k_{share}}||^2 \tag{4}$$

$$\text{s.t.} \quad \boldsymbol{w}^* \in \arg\min_{\boldsymbol{w} \in \mathbb{R}^d} G\left(F_1(\boldsymbol{w}, \mathcal{A}), ..., F_K(\boldsymbol{w}, \mathcal{A})\right), \tag{5}$$

where local model's weights $\boldsymbol{v}_k$ are regularized towards the global model $\boldsymbol{w}^*_{k_{share}}$ via $l_2$ norm regularization, where $\boldsymbol{w}^*_{k_{share}}$ are the weight parameters of the operation set space of $\mathcal{A}$ overlapping (sharing) with $\mathcal{A}_k$ for client $k$. Note that, now, only $\boldsymbol{v}_k$ needs to be optimized in Equation 4, while $\mathcal{A}_k$ is fixed during the optimization. We summarize the pseudo-code of the optimization procedure as SPIDER-Trainer in Algorithm 1. At each device $k$, once we receive the aggregated global model $\boldsymbol{w}^*$, we calculate $\boldsymbol{w}^*_{k_{share}}$ (Line #12 and Line #14). Next, we solve the local sub-problem of $G(\cdot)$ approximately, that is $F_k(\boldsymbol{w}, \mathcal{A})$ (Line #17). Further, each client $k$ solves its local objective given in Equation 4 (Line #18). The updates to $\boldsymbol{w}^*$ are computed the same way as is performed in the standard federated setting, such as FedAvg (McMahan et al., 2017) (Line #6).

**(1) Enabling regularization between an arbitrary personalized architecture and the global model** SPIDER-Trainer connects each personalized model with the global model by enabling the regularization between two different architectures: an arbitrary personalized architecture for the local model $\mathcal{A}_k$ and the global model with Supernet $\mathcal{A}$. This is done by weight sharing. $\boldsymbol{w}^*_{k_{share}}$ is used to regularize a subnet ($\mathcal{A}_k$) model parameters $\boldsymbol{v}_k$ towards the global model shared/common parameters $\boldsymbol{w}^*_{k_{share}}$, as shown in Equation 4.

**(2) Avoiding heterogeneous aggregation** SPIDER-Trainer avoids the aggregation of heterogeneous model architectures at the server side. As such, no sophisticated and unstable aggregation methods are required (e.g. knowledge distillation (Lin et al., 2020), clustering (Dudziak et al., 2022) etc.), and it is flexible to use other aggregation methods beyond FedAvg (e.g. Karimireddy et al., 2020; Reddi et al., 2021) to update the global model.

### 4.4 SPIDER-Searcher: Personalizing Architecture

Although SPIDER trainer is able to collaboratively train heterogeneous architectures, manual design of the architecture for each client is impractical or suboptimal. As such, we further add a neural architecture search (NAS) component, SPIDER-Searcher, in Algorithm 1 (Line #10) to adapt $\mathcal{A}_k$ to its local data distribution in a progressive manner. Furthermore, for the architectural evolution of the local model, please note that $\alpha_{ij}$ takes discrete values. Therefore, most NAS works utilize the continuous relaxation of these discrete variables

---

**Algorithm 1 SPIDER Trainer**

---

1: **Initialization:** initialize $c$ number of clients; $\mathcal{A}$ is the global model with model weight parameters $\boldsymbol{w}$; $\mathcal{A}_k$ is the local model with architecture parameters $\alpha_{ij_k}$ and model weight parameters $\boldsymbol{v}_k$ associated with the $k$-th client; $r$ is the total number of rounds; $t_s$ is the number of rounds to start the architecture search; $\tau$ is the recovery periods in the units of rounds; $F_k(.)$ is the cross-entropy loss function calculated at the local data of client $k$.

2: **Server executes:**

3:     **for** each round $t = 0, 1, 2, ..., r - 1$ **do**

4:         **for** each client $k$ **in parallel do**

5:             $\boldsymbol{w}_k^{t+1} \leftarrow \text{ClientLocalSearch}(k, \boldsymbol{w}^t, t)$

6:         $\boldsymbol{w}^{t+1} \leftarrow \sum_{k=1}^{K} \frac{N_k}{N} \boldsymbol{w}_k^{t+1}$

7:

8:     *function* **ClientLocalSearch**$(k, w^t, t)$: *// Run on client $k$*

9:         Set $\boldsymbol{w}^t$ as $\boldsymbol{w}_k^t$

10:       Search Local model: $\mathcal{A}_k^{t+1} = $ **SPIDER-Searcher**$(\mathcal{A}_k^t, t_s, \tau, t)$

11:       $\boldsymbol{w}_{k_{\text{share}}}^{t+1} = [] \quad$ // empty set

12:       **for** each $\alpha_{ij}$ in $\mathcal{A}_k^{t+1}$ **do**

13:           **if** $\alpha_{ij}$ is nonzero **then**

14:             $\boldsymbol{w}_{k_{\text{share}}}^{t+1} = \text{Append}(\boldsymbol{w}_{k_{\text{share}}}^{t+1}; \boldsymbol{w}_{ij}^t)$ // Append only those weight parameters of $\boldsymbol{w}^t$ which have an overlapping operation set space (edge $i$ and operation $j$) between $\mathcal{A}$ and $\mathcal{A}_k$ architectures

15:       **for** each epoch in $p$ **do**

16:           **for** minibatch in training data **do**

17:             Update Global model: $\boldsymbol{w}_k^{t+1} = \boldsymbol{w}_k^t - \eta_w \nabla_w F_k^{\text{tr}}(\boldsymbol{w}_k^t, \mathcal{A})$

18:             Update Local Model: $\boldsymbol{v}_k^{t+1} = \boldsymbol{v}_k^t - \eta_v \left( \nabla_v F_k^{\text{tr}}(\boldsymbol{v}_k^t, \mathcal{A}_k^{t+1}) + \lambda(\boldsymbol{v}_k^t - \boldsymbol{w}_{k_{\text{share}}}^{t+1}) \right)$

19:       **return** $\boldsymbol{w}_k^{t+1}$ to server

---

to ensure differentiability of the $\alpha_{ij}$ parameters (Liu et al., 2019; He et al., 2020d). However, SPIDER employs the Pertubation-based SPIDER-Searcher to learn $\alpha_{ij}$ in a way that only requires evaluation-based search rather than training-based search (optimizing $\alpha_{ij}$). As shown by (Wang et al., 2021), the evaluation-based search can avoid suboptimal architecture often caused by differentiable NAS. We now present the details of the SPIDER-Searcher.

**Progressive Neural Architecture Search** SPIDER-Searcher dynamically changes the architecture of $\mathcal{A}_k$ during the federated training process. This is feasible because the weight sharing-based regularization can handle an arbitrary personalized architecture (introduced in Section 4.3). Due to this characteristic, SPIDER-Searcher can search $\mathcal{A}_k$ in a progressive manner (shown in Figure 1): **Phase 1**: At the beginning, $\mathcal{A}_k$ is same as the Supernet $\mathcal{A}$ in architecture design. The intention of SPIDER-Searcher in this phase is to warm up the training of the initial $\mathcal{A}_k$ on the client's distribution, therefore, the architecture $\mathcal{A}_k$ does not change for some initial federated rounds; **Phase 2**: After warming up the $\mathcal{A}_k$, SPIDER-Searcher performs edge-by-edge search gradually. In each edge search, only the operation with the highest impact on the accuracy is kept. It also uses a few rounds of training as a recovery time before proceeding to the next round of edge search. This process continues until all edges finish searching; **Phase 3**: After all edges finish searching, SPIDER-Searcher does not change the client's architecture $\mathcal{A}_k$. This serves as a final training of the searched architecture-personalized model. This three-phase procedure is summarized as Algorithm 2. Now, we proceed to elaborate on how we calculate the impact of an operation on the Supernet.

**Operation-level perturbation-based selection** In Phase 2, we specify selecting the operation with the highest impact using operation-level perturbation. More specially, instead of optimizing the mixed operation architecture parameters $\alpha$ using another bi-level optimization as DARTS (also known as gradient-based architecture search) (Liu et al., 2019) to pick optimal operation according to the magnitude of $\alpha$ parameters (*magnitude-based selection*), we assign a constant value to $\alpha$ and use *the impact of an operation on the local validation accuracy* (perturbation) as a criterion to search on the edge. We use this simplified method as it is much more efficient given that it only requires evaluation-based search rather than training-based search (optimizing $\alpha_{ij}$). Further, this method avoids suboptimal architecture (Wang et al., 2021) lead by magnitude-based selection in differentiable NAS.

---

**Algorithm 2 SPIDER-Searcher**

---

1: **Search Space:** $\mathcal{A}_k$ is the local model with architecture parameters $\alpha_{ij_k}$ associated with the $k$-th client; $\mathcal{E}$ is the superset of all edges $\{1, ..., E\}$; $\mathcal{E}_s$ is the remaining subset of edges that have not been searched yet; and each edge $e$ has multiple operations $\{1, ..., o\}$; $t_s$ is the number of rounds to start the architecture search; $\tau$ is the recovery periods in the units of rounds; $t$ is the current round number
2: *function* **ProgressiveNAS**($\mathcal{A}_k^t$, $t_s$, $\tau$, t)
3:    **if** $t \geq t_s$ and ($t$ is multiple of $\tau$) and $|\mathcal{E}_s| > 0$ **then**
4:       $i = $ RANDOM $(\mathcal{E}_s)$ // random selection
5:       // searching without training
6:       **for all** operation $j$ on edge $i$ **do**
7:          evaluate validation accuracy of $\mathcal{A}_k$ when operation $\alpha_{ij}$ is set to zero (removed)($\text{ACC}_{\backslash \alpha_{ij}}$)
8:       for the selected edge $i$, select operation $j$ for which ($\text{ACC}_{\backslash \alpha_{ij}}$) is highest
9:       update architecture $\mathcal{A}_k^{t+1}$: set $\alpha_{ij} = 1$ and $\alpha_{il} = 0 \; \forall l$ (where $j \neq l$), remove $i$ from $\mathcal{E}_s$
10:   **else**
11:      return $\mathcal{A}_k^t$ directly
12:   **return** updated $\mathcal{A}_k^{t+1}$ after selection

---

# 5 Experiments

This section presents the experimental results of the proposed method. All our experiments are based on non-IID data distribution among clients. To generate this non-IID data distribution across clients, we employed the widely used latent Dirichlet Distribution (LDA) (He et al., 2020c; Yurochkin et al., 2019).

## 5.1 Experimental Setup

**Tasks and Datasets** We perform an image classification task on three well-known datasets, **CIFAR10**, **CIFAR100** and **CINIC10**. CIFAR10 dataset (Krizhevsky et al., 2009) consists of 60000 32x32 color images in 10 classes, with 6000 images per class, and CIFAR100 dataset (Krizhevsky et al., 2009) consists of 60,000 images in 100 classes, with 600 images per class. CIFAR100 has more classes and comparatively fewer data per class, therefore, it is considered more challenging than CIFAR10. In addition, CINIC10 consists of 270,000 32x32 color images in 10 classes, with 90,000 images per train, test, and validation subset. CINIC10 dataset (Darlow et al., 2018) is a larger dataset and includes images from ImageNet as well. We generate non-IID data across 8 clients by exploiting LDA distribution with parameter ($\alpha = 0.2$) for the training data of CIFAR10, CIFAR100, and CINIC10 datasets. Since we need validation data for SPIDER-Searcher, we split the total training data samples present at each client into training (60%), validation (20%), and testing sets (20%). We use this 60/20/20% train/valid/test split during Phase 1 and 2 of the SPIDER Training. Once each client has selected the architecture, they combine the validation data with training data in Phase 3 and use it for training. The test set remains the same throughout the training. For all other personalization schemes used for comparison, we split the data samples of each client with 80 % training and 20 % test for a fair comparison. We provide the LDA data distribution class-split, Hyper-Parameter Search Set, and Architecture Search Space details in Appendix A.

**Implementation and Deployment** We implement the proposed method for distributed computing with nine nodes, each equipped with an NVIDIA RTX 2080Ti GPU card. We set this as a cross-silo FL setting with one node representing the server and eight nodes representing the clients. These client nodes can represent real-world organizations such as hospitals and clinics that aim to collaboratively search for personalized architectures for local benefits such as higher accuracy in a privacy-preserving FL manner. Primarily, we focus on the cross-silo setting where clients have ample computational resources and data heterogeneity is a main concern (Huang et al., 2022). However, we provide an analysis of the computational cost of SPIDER in the Appendix E.

## 5.2 Results

Here, we report the comparison of our proposed method SPIDER with the other state-of-the-art personalized methods; `Ditto`, `perFedAvg`, `local adaptation`, `KNN-Per` and `FedMN`. Since these schemes use a pre-defined architecture, we use the Resnet18 model because of its comparable model size. Since we are exploiting DARTS-based search space, we also use a DARTS model (Liu et al., 2019) searched on CIFAR10 dataset as our base model for l2 regularization-based personalization method Ditto. Furthermore, we also provide a comparison with Centralized NAS and Global NAS (FedNAS) in Appendix D.

Table 1: Average (local) test Accuracy and fairness (standard deviation across clients) comparison of SPIDER with the representative state-of-the-art personalization techniques on CIFAR10, CIFAR100, and CINIC10 Datasets

| Method | CIFAR10 | | | CIFAR100 | | | CINIC10 | | |
|---|---|---|---|---|---|---|---|---|---|
| | Average Accuracy | Parameter Size | Standard Deviation | Average Accuracy | Parameter Size | Standard Deviation | Average Accuracy | Parameter Size | Standard Deviation |
| Local Adaptation – ResNet18 | 90.39±0.17 | 11M | 1.9 | 60.92±0.96 | 11M | 2.9 | 80.96±0.96 | 11M | 5.5 |
| Ditto – ResNet18 | 88.91±0.43 | 11M | 2.7 | 61.06±0.23 | 11M | **2.4** | 80.42±0.73 | 11M | 5.5 |
| perFedAvg – ResNet18 | 90.94±0.43 | 11M | 1.6 | 54.56±0.23 | 11M | 4.7 | 79.9±0.28 | 11M | 5.4 |
| Ditto – DARTS | 89.82±0.21 | 3.3M | 1.9 | 60.18±0.27 | 3.3M | 2.3 | 83.61±0.28 | 3.3M | 6.4 |
| KNN-Per ResNet18 | 91.26±0.91 | 11M | 1.6 | 67.05±0.41 | 11M | 2.5 | 85.66±0.39 | 11M | 5.8 |
| FedMN ResNet18 | 88.48±0.25 | 11M | 2.1 | 61.92±0.44 | 11M | 3.4 | 81.41±0.61 | 11M | 6.7 |
| SPIDER | **94.55±0.12** | **2.4M** | **1.5** | **72.36±0.07** | **2.3M** | 2.4 | **87.25±0.65** | **2.1M** | **3.5** |

### 5.2.1 Average Test Accuracy and Fairness (Standard Deviation across Clients)

As shown in Table 1, our proposed method, SPIDER, successfully achieves the objective **R1** by personalizing architectures for each client, surpassing representative state-of-the-art personalization methods such as `Ditto`, `local adaptation`, `perFedAvg`, `FedMN`, and `KNN-Per` on three image classification datasets: CIFAR10, CIFAR100, and CINIC10. Among these methods, `KNN-Per` obtains the second-highest accuracy for all three datasets. However, SPIDER outperforms `KNN-Per` by a margin of 3.29%, 5.31%, and 1.59% on CIFAR10, CIFAR100, and CINIC10, respectively. Moreover, the performance gain of SPIDER over the hypernetwork-based architecture search method, `FedMN`, is quite substantial, reaching 6.10%, 11.31%, and 5.52% on CIFAR10, CIFAR100, and CINIC10, respectively. We attribute this improvement to SPIDER's ability to adapt to unseen data more effectively by tailoring the entire architecture for each individual client based on their specific data distribution.

In the context of personalization, the standard deviation (std) of the local test accuracy across different silos is also an important metric, as it reflects the fairness of the method across silos. In Table 1, we observe that SPIDER achieves similar or lower standard deviation compared to other baselines while providing higher average test accuracy. This demonstrates the effectiveness of SPIDER in achieving both improved average accuracy and lower disparity across silos.

### 5.2.2 Architecture Heterogeneity and Personalization Gain

SPIDER helps each client search for its own architecture tailored for its own specific data distribution. Due to data heterogeneity across clients, we observe architectures to be heterogeneous across clients as shown in Figure 2. Essentially, we visualize the reduction cells searched for CIFAR100 in Figure 2. It can be seen from Figure 2 that the searched cells are edge-wise and operation-wise heterogeneous from client to client. Hence, we achieve the objective **R2**, the search and training of heterogeneous architectures across clients. The objective **R3**, no dependence on auxiliary data, is obtained by the algorithmic framework of SPIDER as it does not rely on any auxiliary data. A byproduct of using DARTS search space is that the searched local model size obtained with SPIDER is quite smaller.

To further investigate whether the architecture searched by one client is best suited for its own data distribution, we perform experiments where the final architecture of one client is applied to another client's data, and the architecture is fine-tuned on that data. We denote the local accuracy obtained on client $i$'s data using architecture $j$ as $p_{ij}$. We calculate the personalization gain or drop (if negative) of other clients'

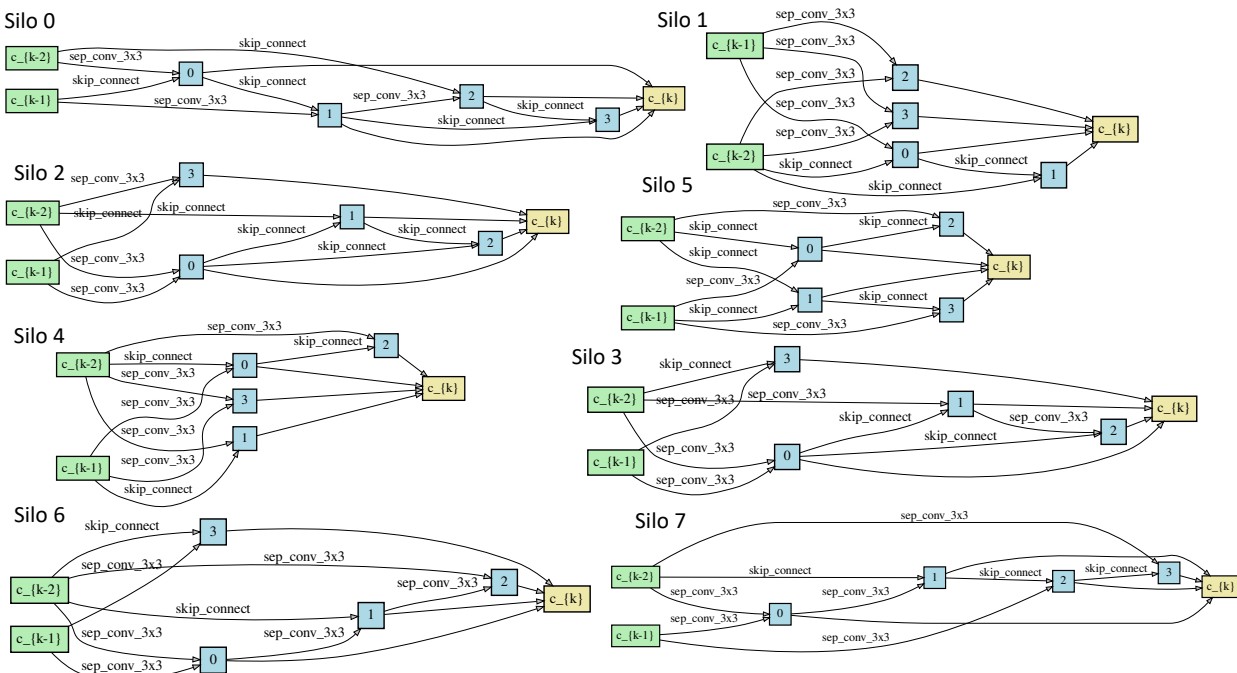

*Figure 2: Searched Architectures (Reduction Cells) for CIFAR100 Dataset: Each normal cell k takes the outputs of previous cells, cell k−2 and cell k−1, as its input. Each cell contains seven nodes: two input nodes, one output node, and four intermediate nodes inside the cell. The output node concatenates all intermediate nodes' output depth-wise. It can be observed that the searched cells are edge-wise and operation-wise heterogeneous from client to client.*

architectures on client $i$'s data using the following formula:

$$\boldsymbol{g}_i = \frac{\sum_{j=0, i \neq j}^{c-1} (p_{ii} - p_{ij})}{c - 1} \tag{6}$$

This calculation is performed across all silos. The quantity $\boldsymbol{g}_i$ represents the personalization gain of architecture $i$ compared to other clients' architectures on its own dataset. We further calculate the personalization gain of the SPIDER scheme as the mean of all $\boldsymbol{g}_i$, i.e., $\frac{\sum_{i=0}^{c-1} \boldsymbol{g}_i}{c}$, where $c$ is the total number of clients. We fine-tune the architecture learned through SPIDER on other clients' data for 30 epochs and report the best accuracy achieved. The values of $p_{ij}$ and $g_i$ for all $i$ and $j$ are reported in Figure 3. As shown in Figures 3a, 3b, and 3c, we observe that for all three datasets, the $P$ matrix is diagonally dominant, indicating that for the majority of clients, their searched architectures outperform the other architectures. Similarly, the personalization gain $\boldsymbol{g}_i$ of each client $i$ is positive, as shown in Figures 3d, 3e, and 3f. Averaged over all clients, the average personalization gains on CIFAR10, CIFAR100, and CINIC10 datasets are 4.70%, 7.84%, and 4.19%, respectively. This highlights the importance of architecture personalization, which can be potentially more effective than weight personalization alone, as demonstrated by our empirical results.

## 6    Conclusion

We proposed SPIDER, an algorithmic framework that aims to search personalized neural architectures for each client in Federated Learning. SPIDER uses progressive NAS to search personalized local architectures and weight-sharing-based global regularization to exchange information between the architecture-homogeneous global and architecture-heterogeneous local models. Our experimental results demonstrate promising prediction performance compared with other state-of-the-art personalization methods.

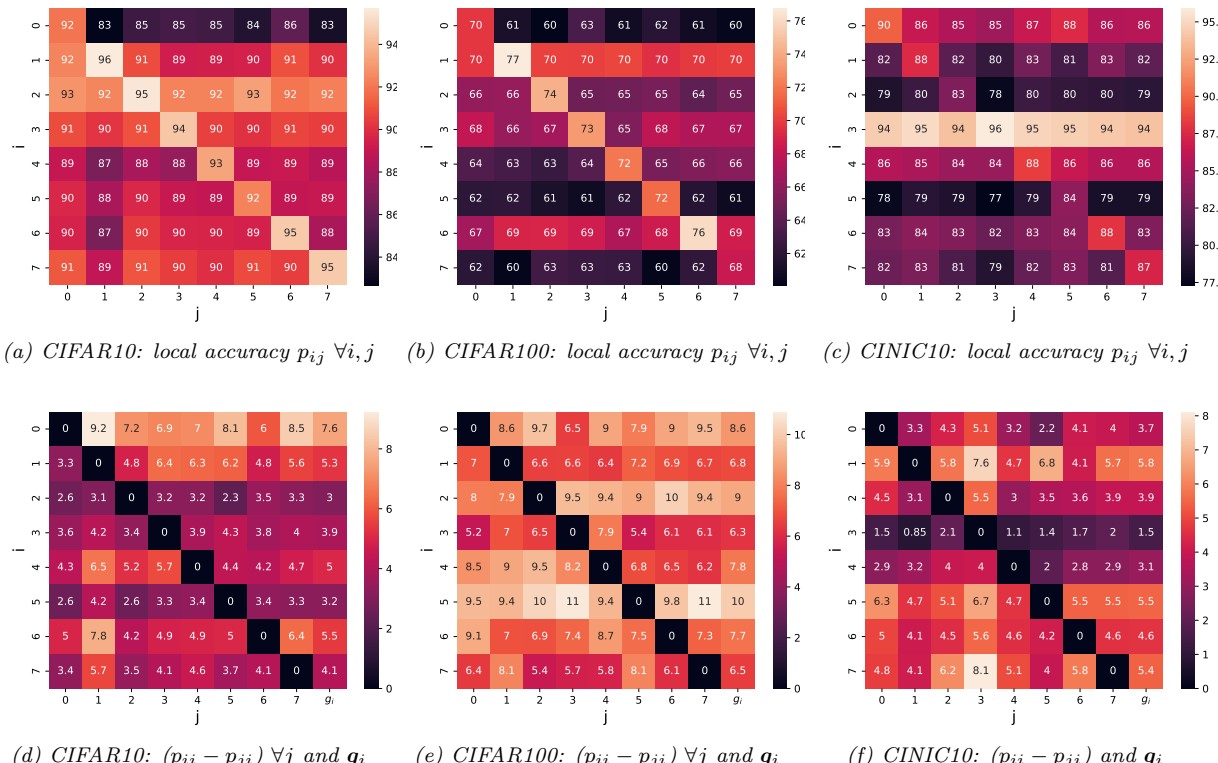

(a) CIFAR10: local accuracy $p_{ij}$ $\forall i, j$     (b) CIFAR100: local accuracy $p_{ij}$ $\forall i, j$     (c) CINIC10: local accuracy $p_{ij}$ $\forall i, j$

(d) CIFAR10: $(p_{ii} - p_{ji})$ $\forall j$ and $\boldsymbol{g}_i$     (e) CIFAR100: $(p_{ii} - p_{ji})$ $\forall j$ and $\boldsymbol{g}_i$     (f) CINIC10: $(p_{ii} - p_{ji})$ and $\boldsymbol{g}_i$

Figure 3: Figures 3a, 3b and 3c represent local accuracy values $p_{ij}$, obtained by finetuning client $j$'s architecture on client $i$'s data on CIFAR10, CIFAR100 and CINIC10 datasets, respectively. Figures 3d, 3e and 3f represent the accuracy gain $(p_{ii} - p_{ij})$ $\forall j$ and the resultant personalization gain values $(\boldsymbol{g}_i)$ for client $i$ obtained on CIFAR10, CIFAR100 and CINIC10 datasets, respectively.

**Future Works:** While the SPIDER framework exploits NAS as a tool to personalize the client architectures and architecture weight parameters in Federated Learning, there are several limitations worth further investigation. Future studies could 1) extend SPIDER to cross-device FL by making it compute-efficient, 2) examine hardware heterogeneity with the data heterogeneity challenge of FL in the context of architecture personalization, and 3) investigate other modality tasks such as text-modeling.

## Acknowledgements

This material is based upon work supported by ONR grant N00014-23-1-2191, ARO grant W911NF-22-1-0165, NSF grant DMS-2134148, and gifts from Intel, Qualcomm, and Konica Minolta.

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

# A  Appendix

# A  Data Heterogeneity

## A.1  Data Heterogeneity

Figure 4 and 5 represents data distribution across 8 clients for CIFAR10 and CIFAR100 datasets, respectively. Sub-figure 4a and 5a represent the label distribution across clients, where a darker color indicates more images of that class/label. It can be seen from these sub-figures that the data distribution is heterogeneous label-wise, e.g, Client with ID 0 has class 0 and 9 in excess, whereas, Client with ID 7 has more samples of class 3 and 7 for the CIFAR10 dataset. We observe a similar trend of heterogeneity for CIFAR100 as well, where we have 100 class labels distributed across 8 clients via LDA distribution with $\alpha$ parameter = 0.2. Furthermore, the sub-figures 4b and 5b represent the total number of data samples preset at each client. These figures illustrate that the number of samples is also varying across silos. However, the variation in the total number of samples by silo is more prominent for the CIFAR10 dataset.

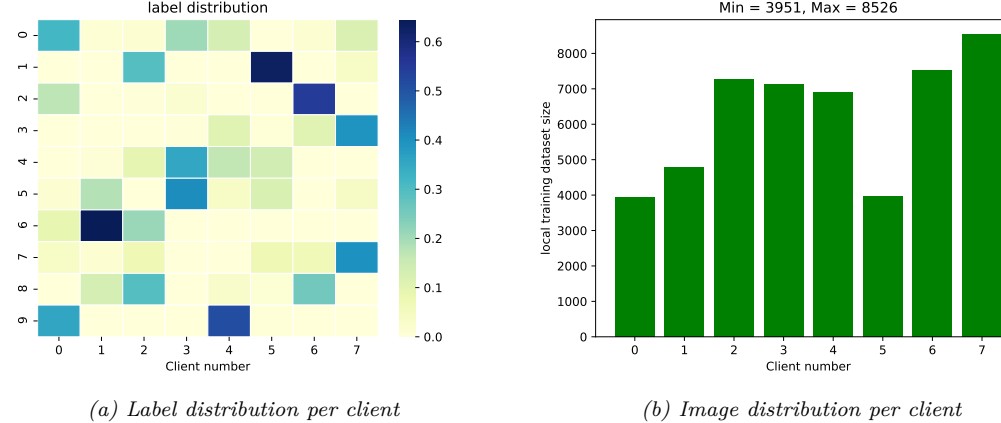

(a) Label distribution per client    (b) Image distribution per client

*Figure 4: CIFAR10: LDA distribution ($\alpha$=0.2) across 8 clients (Seed 9). We can see that the data distribution is heterogeneous in both label and image distribution across clients.*

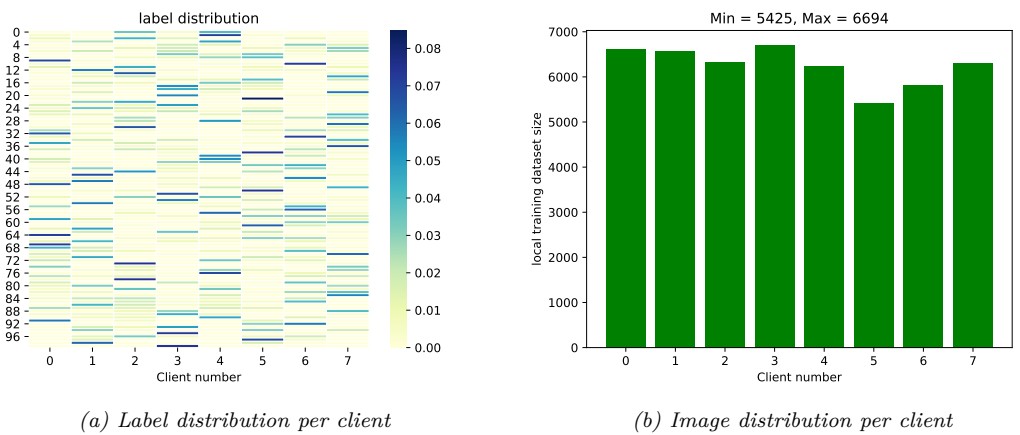

(a) Label distribution per client    (b) Image distribution per client

*Figure 5: CIFAR100: LDA distribution across 8 clients (Seed 9). We can see that the data distribution is heterogeneous in both label and image distribution across clients.*

# B  Hyper-Parameter and Architecture Search Space

## B.1  Hyper-parameter Search

For empirical results of CIFAR10, CIFAR100, and CINIC10, we use a batch size of 32 for all our experiments. We use LDA distribution with a $0.2$ $\alpha$ parameter value. We conduct experiments with two seeds for all methods and report the average values in Table 1. For SPIDER, we use the first 30 rounds as warmup rounds, for SPIDER-Searcher, we use a recovery period of 20. Furthermore, we use a learning rate in the search range of $\{0.01, 0.03\}$ for SPIDER. For SPIDER, we used $\lambda$ search from the set of $\{0.01, 0.1, 1\}$. For the other personalized schemes such as Ditto, perFedAvg, KNN-Per, FedMN, and local adaptation with Resnet18, we searched learning rate over the set $\{0.1, 0.3, 0.01, 0.03, 0.001, 0.003\}$. The reason for having a larger set of learning rates for these methods is that we found 0.001 and 0.003 work better for some of these methods. For Ditto, we used $\lambda$ from the set $\{0.01, 0.1, 0.5, 1, 2\}$. For PerFedAvg, we used the local lr as a factor of $\{1, 3, 5\}$ of the global lr. We used 1000 rounds of communication for the reported results and observed that they were sufficient to achieve convergence as shown in Figure 6. For FedMN, we used 500 rounds of communication as pretraining. After pretraining, the next 500 for federated local training. We report the best average local test accuracy during the federated local training phase in Table 1. For KNN-Per, we evaluate the global model on the $\lambda$ hyper-parameter selected from the set $\{0, 0.1, 0.2, 0.3, 0.4, 0.5, 0.6, 0.7, 0.8, 0.9, 1\}$ and report the best average local test accuracy in Table 1. We used stochastic gradient descent (SGD) optimizer for all the methods.

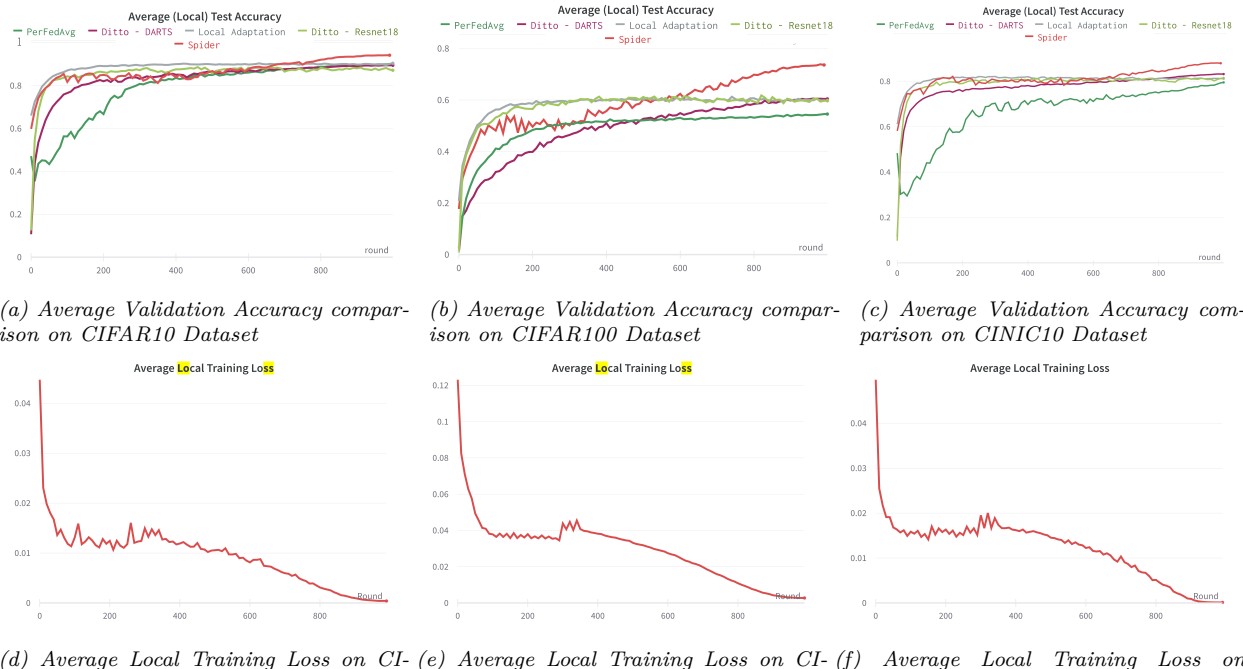

*(a) Average Validation Accuracy comparison on CIFAR10 Dataset*

*(b) Average Validation Accuracy comparison on CIFAR100 Dataset*

*(c) Average Validation Accuracy comparison on CINIC10 Dataset*

*(d) Average Local Training Loss on CIFAR10 Dataset*

*(e) Average Local Training Loss on CIFAR100 Dataset*

*(f) Average Local Training Loss on CINIC10 Dataset*

*Figure 6: Figure 6a, 6b, 6c represent the average validation accuracy comparison between SPIDER and PerFedAvg, Ditto, and Local Adaptation on CIFAR10, CIFAR100, and CINIC10 datasets, respectively. SPIDER outperforms the representative baselines on all three datasets. Figures 6d, 6e and 6f illustrate the change in Average local training loss of the local models as SPIDER Search progresses through three phases (Phase 1 (Local Architecture Pre-training), Phase 2 (Local Architecture Search), Phase 3 (Local Architecture Training)) on CIFAR10, CIFAR100, and CINIC10 Datasets, respectively*

## B.2 Architecture Search Space

As mentioned in Section 4.2, we have used DARTS (s2) search space (Wang et al., 2021) in our proposed work. During the search, we are using a total of 8 cells as shown by Figure 7. Each type of cell consists of 14 edges; each edge connects two intermediate representations (node) by a mixture of two operations frequently used in various modern CNNs, sep convolution 3x3 and skip connection, hence the name 's2 DARTS search space' (Wang et al., 2021). In this search space, there are two types of cells: normal and reduction cells. Hence, our supernet $\mathcal{A}$ consists of search space of normal cells and reduction cells, each with 14 edges and 2 operations at each edge. As the search progresses via progressive NAS, some operations and edges are removed based on the perturbation criterion as explained in Algorithm 2. It is important to note that the same cells follow the same construction; therefore, if one operation is selected for one normal cell at a particular edge, the same operation will be selected for the same edge at all normal cells. More specifically, we have 14 learnable edges and 2 operations at each edge in a cell, there are a total of $2^{14}$ possible configurations. Since we have two types of cells, a normal cell, and a reduction cell, there are a total of $(2^{14})^2$ possible architecture designs. The other DARTS search spaces, s3 and s4 (Wang et al., 2021) have a larger search space, e.g., a larger operation candidate set, and therefore, require more computational resources.

## C  Gradient-based NAS as SPIDER-Searcher

SPIDER is formulated as a bilevel optimization problem as shown in Equation 2 and 3, where the lower-level optimization optimizes the global model parameters $w^*$, and the upper-level optimization optimizes the local architecture $\mathcal{A}_k$ and its weight parameters $v_k$ for a client $k$. For the proposed lower-level optimization of SPIDER, we exploit the perturbation-based neural architecture search method as it simplifies the architecture search process by selecting architectures based on the evaluation criterion only. This searcher picks optimal operations based on the impact of its absence (a.k.a perturbation) on the local validation accuracy as explained in detail in Algorithm 2. On the other hand, differentiable NAS is another widely known method of architecture selection which is often formulated in itself as a bilevel optimization problem with architecture parameters $\alpha$ as an upper-level variable and $w$ as lower-level variables. As an ablation study, we replace perturbation-based NAS Searcher with MileNAS Searcher (He et al., 2020d) which is a gradient-based NAS Searcher. We use its first-order approximation which essentially applies softmax operation to the architecture parameters $\alpha$ for each edge and updates them by the stochastic gradient step $\mathcal{A}_k^{t+1} = \mathcal{A}_k^t - \eta_\alpha \left( \nabla_\alpha F^{\mathrm{tr}}(v_k, \mathcal{A}_k^t) + \lambda_\alpha \nabla_\alpha F_{\mathrm{val}}(v_k, \mathcal{A}_k^t) \right)$ at each iteration of local epochs. $F^{\mathrm{tr}}$ and $F^{\mathrm{val}}$ are calculated at the training and validation data, respectively. For simplicity, $\mathcal{A}_k$ here represents a collection of all architecture parameters in the local model of client $k$, and $\eta_\alpha$ is the learning rate of architecture parameters. We select $\eta_\alpha$ to be 0.01 and the regularization parameter $\lambda_\alpha$ as 1. We apply MileNAS solver for the first 400 communication rounds and find the local architectures based on the magnitude of the architecture parameters $\alpha$ at each edge, and then train them for the next 600 rounds by following SPIDER formulation given in Equation 4 and 5. As shown in Table 2, the performance gap between the two methods is negligible for the CIFAR100 dataset. However, perturbation-based NAS is a simpler method as it does not require a gradient step for the architecture search/update at each step, and therefore, takes less amount of time in comparison to gradient-based NAS.

*Table 2: Average (local) test Accuracy comparison of Perturbation-based NAS with gradient-based NAS in SPIDER on CIFAR100 dataset.*

| Method | Average Accuracy |
|---|---|
| Perturbation-based NAS  (Wang et al., 2021) | 72.36 |
| MiLeNAS  (He et al., 2020d) | 72.40 |

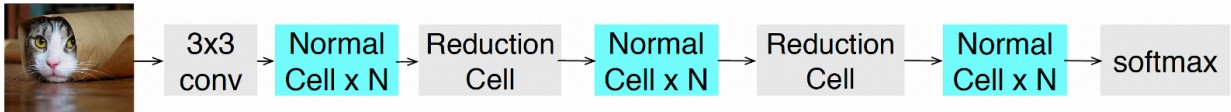

*Figure 7: Supernet $\mathcal{A}$ search space. Note that since we have 8 cell based search space, N = 2 for our construction.*

# D    Federated Neural Architecture Search versus SPIDER

In this section, we compare the proposed Personalized Neural Architecture search method SPIDER with its two base cases, centralized learning where $\lambda = 0$ and architecture personalization base case where $\lambda \to \infty$ which essentially refers to the setting where local model weights are identical to the global model weights. We also compare SPIDER with the federated neural architecture search method, which searches a global architecture in a federated manner and then trains it via FedAvg. This investigation can help us appreciate the benefits of architecture personalization and federated learning in a cross-silo setting.

## D.1    Federated Neural Architecture Search

We analyze NAS performance to search for a global architecture in a federated setting as proposed by previous works (He et al., 2020b). To keep the algorithmic comparison fair, we have used the NAS setting as same as SPIDER. Essentially, only one Supernet (DARTS) is maintained at each silo. Each silo trains the Supernet following the FedAvg algorithm for 30 rounds of warmup. Next, the neural architecture search is performed following perturbation-based NAS at the server using the server's test/validation data. After NAS, the architecture is obtained and trained in a federated manner by the FedAvg algorithm. This can be considered a slight variation of FedNAS (He et al., 2020b) where we replace MiLeNAS He et al. (2020d) with a state-of-the-art NAS method, Perturbation-based NAS (Wang et al., 2021) to keep the comparison fair. Figure 8 represents the average validation accuracy across each silo. We obtained a total of 68.03% average local test accuracy with global NAS. Although this accuracy is still higher than various PFL baselines, it is lower than SPIDER (72.36%). This shows that a global architecture search may not capture the data heterogeneity in FL with non-I.I.D data.

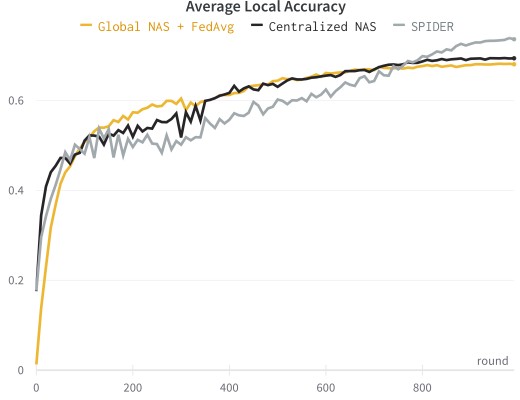

*Figure 8: Average Validation Accuracy comparison of SPIDER with the Federated NAS on CIFAR100 dataset. For SPIDER, we show the results of two different values of $\lambda$ hyper-parameter. First setup of SPIDER is Centralized NAS where $\lambda = 0$. The second setting of SPIDER is $\lambda = 0.01$, which outperforms the centralized NAS, a special case of SPIDER ($\lambda = 0$) as well as Federated NAS.*

## D.2    SPIDER: Personalized Federated Neural Architecture Search

We can observe from Figure 8 that SPIDER with $\lambda = 0.01$ hyper-parameter setting outperforms the Federated NAS. However, it is important to explore two base cases of SPIDER. The first setting is where $\lambda = 0$ and therefore, has no learning between the local and global models. It is where each silo performs a local

neural architecture search. Each silo uses only one model, Supernet, and deploys the perturbation-based NAS Searcher locally **without any collaboration from the other silos**. The Local Supernet has the same hyper-parameters of warmup (30 local epochs) and recovery period of 20 local epochs after every pruning step, followed by local training with a total of 1000 epochs. We obtained 72.36% average local test accuracy with SPIDER, and 69.31% average local test accuracy with Centralized NAS on CIFAR100 dataset. This highlights the significance of the proposed regularization. This also shows that tailoring architectures to the silo's data distributions even for centralized training can be very powerful. However, collaboration with other silos as is performed by SPIDER, can enhance the prediction performance of personalized architectures at individual silos.

## E    Computational Cost Analysis of Training

We propose SPIDER as a cross-silo framework where clients have ample computational resources and data heterogeneity is a main concern (Huang et al., 2022). As shown in Table 3, we achieve substantial performance improvement in terms of average local accuracy over the state-of-the-art personalization methods. For research purposes, it is essential to compare the computational cost and wall clock time of the proposed method with the other representative personalized federated learning (PFL) methods. Therefore, we report peak memory costs and total wall clock time for all methods. Note that we obtain the peak memory cost values for the maximum number of compute required for the model training, which includes forward/backward pass size, and parameter size of all the models, global and local model, for a batch size of 32 on the NVIDIA RTX 2080Ti GPU card.

We observe the peak compute cost per silo to be 4.18GB, 2.14GB, 1.45GB, 1.45GB, 2.90GB, 1.45GB, and 1.98GB for SPIDER, FedNAS, Local adaptation, KNN-Per, Ditto, perFedAvg, and FedMN, respectively. Though SPIDER requires relatively higher peak memory for maintaining two models, a global supernet and a local model, at each silo, it outperforms the state-of-the-art personalization methods substantially. For example, SPIDER yields a 6.18% performance gain relative to the second-best method, KNN-Per, on the CIFAR100 dataset which can be a significant gain for organizations that require high accuracy without the constraint of small computing cost.

Further, we report the end-to-end time of each method for a total of 1,000 rounds on the CIFAR100 dataset. For implementation, we used 8 GPU NVIDIA RTX 2080Ti GPU cards where each GPU represents a physical node with an NVIDIA RTX 2080Ti GPU card on the fedml platform implementation for FL (He et al., 2020c). Please note that to report FedMN and KNN-Per method results, we used their GitHub code directly where they perform federated learning on one GPU in a sequential manner. Therefore, the time comparisons of these two methods might not be fair. Overall, for wall-clock time comparisons, we observe that there are two factors contributing to the wall-clock overhead.

**Overhead of Neural Architecture Search (NAS):**    One notable observation is that methods that use a predefined architecture tend to have significantly lower wall clock times. However, this comes at the cost of lower prediction performance. It is important to note that these low wall clock times may not hold true in practical scenarios, especially when dealing with the data-invisibility challenge in Federated Learning (FL).

The data invisibility challenge in FL refers to the distributed learning setup where clients' private data remain invisible to the server. As a result, from the server's perspective, it becomes unclear how to select a pre-defined architecture from a pool of all available candidates. For instance, in our work, we have reported results for the Ditto method using two predefined architectures, resnet18 and DARTS. Considering both architectural experiments, the wall clock time has doubled to 8 hours and is even higher if we include the hyper-parameter search time for both architectures, yet the performance remains significantly lower ( 12% lower than SPIDER).

Given the data heterogeneity and data invisibility challenges of FL, manually searching for a neural architecture that works optimally for all clients can become prohibitively expensive. This challenge has been one of the motivations for developing SPIDER as a personalized neural architecture search framework.

Table 3: Accuracy versus Computational Cost Tradeoff on CIFAR100 dataset for SPIDER versus the other representative personalized federated learning techniques such as Local Adaptation, Ditto, KNN-Per, perFedAvg, FedMN.

| Method | Accuracy | Peak Memory Cost | Wall Clock Time |
|---|---|---|---|
| SPIDER | 72.36% | 4.18GB | 15 hour |
| FedNAS | 68.03% | 2.14GB | 7 hour |
| Local Adaptation - ResNet18 | 60.92% | 1.45GB | 2 hour |
| KNN-Per - ResNet18 | 67.05% | 1.45GB | 10 hour |
| Ditto - ResNet18 | 61.06% | 2.90GB | 4 hour |
| perFedAvg - ResNet18 | 54.56% | 1.45GB | 5 hour |
| FedMN - ResNet18 | 61.92% | 1.98GB | 16 hour |

**Overhead of Bilevel Optimization:** Another factor contributing to overhead is the Bilevel Optimization where the lower-level optimization optimizes the global model parameters and the upper-level optimization optimizes the local architecture and its weights parameters. For example, we observe the wall clock time of SPIDER to be almost double the wall clock time of FedNAS (NAS-based method that does not deploy bilevel optimization for personalization). However, SPIDER demonstrates approximately a 5% increase in prediction performance. Likewise, we observe the wall clock of Ditto to be double the wall clock time of Local Adaptation.

It is important to highlight that our main focus has been to address the data-heterogeneity challenge, especially in cross-silo FL settings where silos can have higher computational resources and non-IID data distributions are the main challenge. We show via extensive empirical experiments that the proposed method yields higher predictive performance with lower standard deviation (a proxy metric for fairness) across silos as compared to other state-of-the-art methods at the cost of higher computational power.

