# OpenReview forum: "Distributed Architecture Search Over Heterogeneous Distributions"
_TMLR — Accepted by TMLR_

### Review · Reviewer_ZfAU · 2023-08-16

**Summary Of Contributions:**

This paper considers the problem of personalization in federated learning. The authors propose a novel neural architecture search based method, termed SPIDER, to address this problem. More precisely, they
apply general federated learning to sync global model and use a NAS method to search for local models and use ditto regularization to penalize the distance between shared weights and local weights. They use DARTS search space and use validation set to pick the best architecture. The empirical performances suggest that SPIDER outperforms other state-of-the-art methods.

**Audience:**

Yes

**Broader Impact Concerns:**

No.

**Claims And Evidence:**

Yes

**Requested Changes:**

Please address the weaknesses in the previous section.

**Strengths And Weaknesses:**

**Strength**:
- The overall idea is clear and makes sense.
- The empirical performance shows that compared to other state-of-the-art methods SPIDER obtains better accuracy with a smaller model.

**Weakness**:
- It is unclear to me how large is the wall clock overhead of SPIDER compared to other methods.
- The authors did not provide guarantee that such bilevel optimization will converge.
- How does the stochasticity of NAS influence the result?

---

### Review · Reviewer_umJq · 2023-09-11

**Summary Of Contributions:**

This paper presents a method for model personalization in cross-silo federated learning by finding a silo-specific NN architecture and corresponding weight vectors. The authors showcase their method with a range of toy-examples and analyze its performance comparing it with other model personalization methods in FL, including an analysis of std as a proxy for fairness in this setup.

**Audience:**

Yes

**Broader Impact Concerns:**

There are no concerns with ethical implications. I would encourage the authors to consider the case of hardware-heterogeneity and how different resource constraints could impact the NAS procedure as an outlook to potential future work.

**Claims And Evidence:**

No

**Requested Changes:**

In order to recommend acceptance, I would like the authors to discuss the simple baseline of keeping client and server weights identical, similar to the traditional FedAvg setting. Is there a reason why this baseline does not make sense in this setting?
Further, please substantiate or remove the claim about adversarial robustness.

Beyond those, I found the following small issues:
- Figure 3 legend: You reference 2a, 2b and 2c as sub-plots, the numbering is off.
- Figure 2 legend vs text. In the legend you claim Cifar10, in the text you mention Cifar100.

**Strengths And Weaknesses:**

The paper's strength lies in its exhibition. The method is well-motivated and explained along with helpful graphics and algorithm descriptions. The problem is highly relevant and I enjoyed reading the paper.
The paper's weaknesses are as follows:
- Adversarial attacks claim. There is no study of the method's robustness to adversarial attacks and the claim is hand-wavy. Given that the method optimizes $w_k$ locally and communicates to the server, there is ample opportunity to adversarially attack $w$ by a malicious client.
- Missing baseline and opportunity. The authors present their method as a solution to R1 & R2 specifically. It involves not only optimizing $A_k$, but also includes finding localized model weights $v_k$ for those localized architectures. The FL-aspect stems solely from $\frac{\lambda}{2}Reg(v_k,w*)$. I would like to see the baseline in which $\lambda \rightarrow \inf$ and the clients' and server weights are kept identical. Instead of personalizing both, weights and architecture, such a method would be simpler and involve only architecture search as a means of personalization. With a search-space of size $2^{14}$, there should be enough opportunity for personalized results. Such a simpler method would also open up the possibility for cross-device FL and one-shot personalization as a client could perform the perturbation based architecture search upon joining the federation. This method would still satisfy R1-R3 as far as I can tell.
- Missing evaluation opportunity: I would have loved to understand if similarity in data is followed by similarity in the selected model architecture. Do two clients with similar (or even identical) label-distribution 'find' a similar (or identical) architecture? For two clients with identical label distribution, such an outcome would be desirable since it increases the statistical strength of the shared model architecture weights.
- Datasets and settings: While I believe the results to be conclusive given the presented datasets, it would have been a stronger statement for the paper to present a dataset of different modality, scale or task-structure. At the moment, the experiments capture only image classification on low-resolution natural images. The problem definition of FL-NAS however captures a much wider range. I would encourage the authors to consider e.g. a text-modeling task or an object-localization task and dataset.

---

### Review · Reviewer_wBn9 · 2023-09-15

**Summary Of Contributions:**

This paper proposes a method for personalizing the neural architectures for differnt clients in federated learning. The proposed method consists of two phases. During the training phase, the heterogeneous architectures of different clients are collaboratively trained by regularizing their weights using a global architecture model. Then during the searching phase, the personalized architectures of every client is further optimized using a simple perturbation-based heuristic method.The experimental comparison shows that the proposed method which personalizes the neural architecture improves over previous methods which only personalize the model weights.

**Audience:**

Yes

**Broader Impact Concerns:**

I do not see ethical concerns about this paper.

**Claims And Evidence:**

Yes

**Requested Changes:**

Please see the points I've listed under Weaknesses above for the requested clarifications and changes.

**Strengths And Weaknesses:**

Strengths:
- The proposed method is well motivated. That is, most previous methods on personalized FL only aim to personalize the weights of the neural network. This paper goes one step further and additionally personalize the neural architectures, and hence achieves improvements of these previous methods.
- The algorithmic designs are all intuitive and reasonable. The proposed method of regularization to enforce knowledge sharing between the global model and local models is particularly interesting.
- The experiments are nicely done. The results in Table 1 indeed demonstrate improvements over the previous methods. Fig 2 is a nice illustration of the reasonableness of the proposed approach of personalizing the neural architectures for different clients. The design of the personalization gain is also interesting and reasonable.

Weaknesses:
- I think the computational cost of the proposed method should be evaluated or discussed. If I understand correctly, the proposed method of personalizing the neural architectures is much more complicated and hence potentially more time-consuming than the previous methods which only personalize the model weights (e.g., the baseline methods compared in Table 1). I understand that sometimes speding more computation is necessary to achieve improvement in the performance such as those shown in Table 1, but I think comparisons should be made in terms of the computational cost.
- Although there has been some previous works on doing NAS in FL (as reviewed in Section 2), however, the proposed method is only compared with one of these methods (Federated NAS) in Appendix C, and the results show that the proposed method isn't really siginificantly better than FedNAS since the proposed  only starts to outperform FedNAS after a large number of iterations. I think more comparisons with the previous federated NAS methods should be performed, or if comparisons with the previous methods mentioned in Section 2 are not feasible, it should be clearly explained.
- As described in Section 4.4, the local architecture searching phase is performed using a simple perturbation-based heuristic instead of a gradient-based NAS method, even though the proposed method adopts the DARTS search space which was developed for gradient-based NAS. I think it would be nice to see whether the use of gradient-based NAS (such as DARTS) can further improve the local architecture search phase.

---

### Decision · Action_Editor_S8CP · 2023-11-09

**Recommendation:** Accept as is

**Comment:**

The computational overhead of the methods was discussed. Although the proposed method incurs more computational time, the large performance advantage of the proposed method seems to justify the additional computations.

**Audience:**

Yes, methods for personalized federated learning are of interests to the TMLR audience.

**Claims And Evidence:**

This paper presents a neural architecture search based method for model personalization in cross-silo federated learning.

The reviewer found that all claims in the revised version of the paper are supported by clear empirical evidence. The experiments show clear improvements over the implemented baseline methods.